# The Relation between Neuroticism and Non-Suicidal Self-Injury Behavior among College Students: Multiple Mediating Effects of Emotion Regulation and Depression

**DOI:** 10.3390/ijerph19052885

**Published:** 2022-03-02

**Authors:** Chengju Liao, Xingmei Gu, Jie Wang, Kuiliang Li, Xiaoxia Wang, Mengxue Zhao, Zhengzhi Feng

**Affiliations:** 1Department of Medical Psychology, Army Medical University, Chongqing 400038, China; lcj013@163.com (C.L.); risyaiee@msn.cn (K.L.); lemonowang@gmail.com (X.W.); zhao_meng_xue@163.com (M.Z.); 2Department of Medical English, Army Medical University, Chongqing 400038, China; gxm@tmmu.edu.cn; 3Department of Finance and Economics, Chongqing Chemical Industry Vocational College, Chongqing 401228, China; lancywangjie@163.com

**Keywords:** neuroticism, emotion regulation, depression, non-suicidal self-injury, multiple mediating effect

## Abstract

Background: Non-suicidal self-injury (NSSI) behavior among college students is a focus of attention in current society. In the information era, the Internet serves as a public health concern and as an effective pathway for prevention. In order to reduce NSSI behavior, we explore its influence factors, especially the relations between neuroticism, emotion regulation (ER), depression, and NSSI behavior. Methods: A total of 450 college students were surveyed with the Big Five Inventory-2, Emotion Regulation Questionnaire, Self-Rating Depression Scale, and Adolescent Non-Suicidal Self-Injury Assessment Questionnaire. Results: Regression analysis showed that neuroticism significantly negatively predicted emotion regulation, while it positively predicted depression and NSSI. Multiple mediation modeling demonstrated that neuroticism and emotion regulation had no significant direct effects on NSSI. However, neuroticism could indirectly affect NSSI through four pathways of multiple mediating effects, including depression, cognitive reappraisal-depression, expressive suppression-depression, and cognitive reappraisal-expressive suppression-depression. Conclusions: Neuroticism positively predicts depression and NSSI behavior, and affects NSSI through the mediating effect of ER and depression. Therefore, amelioration of neuroticism from the perspectives of emotion regulation and depression is recommended, so as to reduce NSSI behavior among college students with highly neurotic personalities.

## 1. Introduction

Non-suicidal self-injury (NSSI) is defined as the deliberate, self-inflicted destruction and self-punishment of body tissue without suicidal intent and for purposes not socially sanctioned, including behaviors such as cutting, burning, biting, and scratching skin [1,2]. According to the general strain theory (GST) brought forward by Agnew, external stressful events or situations can trigger negative emotions—such as anxiety, depression, anger etc.—and to release those negative emotions, individuals tend to react by attacking others or injuring themselves [3]. In the information era, information on self-injury motivation or methods are easily available online, exposure to such information may increase the risk of NSSI behavior among highly neurotic individuals. In a sample of over 3500 school pupils, O’Connor et al. found that 18% of those who had self-harmed indicated the Internet or social networking sites influenced their decision to engage in self-harm. The Internet has, to a certain degree, prompted NSSI behavior, leading to an increasing incidence of NSSI behavior in recent years [4]. A meta-analysis showed that the aggregate lifetime and 12-month prevalence of NSSI in children and adolescents between 1989 and 2018 were 19.5% and 22.1% respectively [5]. In recent years, the number of reports on NSSI among college students has been increasing. A two-year annual follow-up assessment from the Leuven College Surveys indicated that one-year incidence of first onset NSSI was 10.3% in year 1 and 6.0% in year 2 [6]. An epidemiological study among adolescents and young adults aged 14–21 years illustrated that lifetime NSSI was reported by 19.3% of the sample [7]. NSSI behaviors, being covert and hard to detect, have strong impacts on an individual’s physical and psychological health and significantly predict suicidal ideation and suicidal behavior [8,9]. Recently, NSSI has been included as an independent disorder in the international DSM-5 classification system [10]. Therefore, it is important to reduce NSSI behavior and explore its influence factors.

Factors influencing NSSI are varied, such as traumatic experience and individual susceptibility [11]. If someone was subjected to childhood traumatic experiences—i.e., physical and emotional abuse or neglect during childhood and adolescence—their functional impairments have been reliably established. Such negative incidents lead to individual susceptibility and have a destructive effect on the development of adult personality in the long term. Neuroticism, a critical characteristic of individual susceptibility, significantly predicts NSSI behavior [12]. Highly neurotic individuals are generally more introverted and sensitive. When adverse events occur in their life, they have deviation in processing stimuli (especially negative information) and regard it as a threat, and tend to respond by injuring themselves. Additionally, recent evidence has implicated experiential avoidance (EA) as a potentially important factor in the engagement of NSSI behaviors and suicidal ideation [13,14]. The EA model posits that self-injury is regarded as an avoidance and escape behavior in response to unwanted and distressing cognitions or emotions, or experiences of uncontrollable negative emotional arousal. Engagement in self-injury can narrow attention to physical pain experienced in the here and now which can also alleviate acute distress [15]. In general, an individual with a neurotic personality is more liable to be trapped in the vicious circle of NSSI behavior. However, not all individuals with neurotic personality will react through NSSI behaviors in the face of the same conflict situations. Those indulging in negative feelings are likely to implement NSSI behaviors. Some researchers argued that neuroticism represents the response system of negative emotions and evokes stronger physiological and emotional response in negative emotion-inducing situations [2,16]. If constantly experiencing harassment, slander, or exposure of privacy, highly neurotic individuals are disposed to have negative emotions, such as depression, anxiety, anger, shame, and guilt. When they cannot effectively deal with these emotions, they tend to punish themselves or seek mental stimulation through NSSI. This shows that negative feelings, especially depressive emotion, are an important factor associated with NSSI behavior among neurotic individuals [17,18]. Therefore, alleviating their negative emotions may be considered as a way to reduce or mitigate their NSSI behaviors. 

Emotion regulation (ER) refers to how we try to influence which emotions we have, when we have them, and how we experience and express these emotions. It is one of the most important tools to reduce depression and other negative emotions. Guerin-Marion et al. revealed that more impaired ER functions were associated with more frequent lifetime NSSI [19]. According to the process model of emotion regulation put forward by Gross, cognitive reappraisal and expressive suppression are two common ER strategies [20]. Individuals who tend to use cognitive reappraisal after adverse life events, adopt a more positive attitude towards traumatic events, effectively altering the understanding of emotional events and establishing new understanding of stressful events in good psychological state; while individuals who resort to expressive suppression need to mobilize resources to inhibit the expression of negative emotions when they have them so as to avoid the compulsive interference of negative automatic thoughts. Since ER strategies can regulate and reduce depressive emotion, which is a critical proximal factor leading to NSSI, we therefore hypothesized that ER is closely associated with depression and both ER and depression may be important predictors of NSSI. 

Previous studies have found that highly neurotic personality is associated with enhanced maladaptive regulation of negative emotions [21,22]. Neurotic individuals show top-down maladaptive regulation of negative emotions, causing social anxiety and depression and affecting life satisfaction and mental health. Maladaptive cognitive strategies, such as rumination or catastrophizing, are related to higher levels of loneliness, especially for introverts [23]. Individuals with high maladaptive regulation report lower well-being [24]. Other studies revealed that maladaptive emotion dysregulation (e.g., self-blame, acceptance, rumination, catastrophizing, and other-blame) serve as concurrent mediators of the association between neuroticism and depression severity. In contrast, adaptive cognitive ER strategies (e.g., positive refocusing, refocusing on planning, positive reappraisal, and putting into perspective) have significant, negative correlations with neuroticism and depressive symptoms [25,26]. This demonstrates that neurotic individuals may resort to ER when experiencing negative emotions. These findings provide an important and strong basis for exploring the factors of NSSI behavior. 

Collectively, these findings indicate that highly neurotic individuals tend to adopt maladaptive ER strategies and are prone to depression due to impaired ER. Long-term depression, however, is an important proximal factor causing NSSI, which may eventually lead to NSSI behaviors among individuals. To explore the relationship between neuroticism, ER, depression, and NSSI behavior, we established the following hypotheses: 

**Hypothesis 1** **(H1).**
*Neuroticism significantly negatively predicts ER strategies, and positively predicts depression and NSSI behavior;*


**Hypothesis 2** **(H2).**
*ER and depression have multiple mediating effects on neuroticism and NSSI behavior.*


## 2. Materials and Methods

### 2.1. Participants and Procedure

From 28 May to 5 July 2021, we conducted a questionnaire survey among 500 college students ranged from freshman to senior year in two universities in Chongqing. Each of them filled out the questionnaire independently within 20 min. Excluding questionnaires in which basic information was missing or unfinished or some options were unchosen, and questionnaires of those who had not completed the whole survey, we collected 450 valid questionnaires (response rate 90%). Among these respondents, 202 (44.89%) of them were male and 248 female (55.11%), with an average age of 16 to 25 years (M = 19.48, SD = 1.13).

### 2.2. Research Materials

The Emotion Regulation Questionnaire (ERQ) [20] is commonly used to assess individual differences in the habitual use of two ER strategies: cognitive reappraisal and expressive suppression. It contains 10 items, 6 of which are reappraisal items and 4 of which are suppression items. It uses a seven-point Likert scale scored from 1 (totally disagree) to 7 (totally agree). The higher the score, the stronger the regulation habit. The Cronbach α coefficient of internal consistency was 0.76 and 0.68 for cognitive reappraisal and expressive suppression respectively.

Six items in the Big Five Inventory-2 (BFI-2) [27] were selected to assess neuroticism and a five-point Likert scale scored from 1 (totally disagree) to 5 (totally agree) was adopted. The higher the score, the more consistent with the personal characteristic description. The Cronbach α coefficient of internal consistency was 0.84. 

Depressive symptoms were measured by Self-Rating Depression Scale (SDS) [28], containing 20 items, each of which consists of 4 options, namely “a little of the time, some of the time, a good part of the time, and most of the time” scored by 1–4 points respectively. The standard score = int (Original Score × 1.25), ranging from 25 to 100. Standard score < 50 means no depression, 50–59 mild depression, 60–69 moderate depression, and ≥ 70 major depression. The Cronbach α coefficient of internal consistency was 0.73.

The Adolescent Non-Suicidal Self-Injury Assessment Questionnaire [29] is composed of a behavior questionnaire and function questionnaire. There are 12 items in the behavior questionnaire, including NSSI behaviors with no obvious tissue damage (such as pinching, scratching, hair pulling, etc.) and obvious tissue damage (such as cutting, burning, lettering or symbols on the skin, etc.). Each item contains five options—namely “no, occasionally, sometimes, often, and always”—which are scored by 0–4 points respectively, with a total score of 0–48 points. The higher the score, the higher the frequency of NSSI behavior. There are 19 items in the function questionnaire, which are divided into three dimensions: egoistic social interaction, self-negative reinforcement, and emotional expression. Each item contains five options—namely “completely inconsistent, inconsistent, uncertain, consistent, and fully consistent”—which are scored by 0–4 points respectively, with a total score of 0–76 points. The higher the score, the greater the possibility of conducting NSSI behavior. The Cronbach α coefficient of internal consistency was 0.92 and 0.91 for behavior questionnaire and function questionnaire, respectively.

### 2.3. Statistical Analysis

Firstly, IBM SPSS Statistics 25.0 was used to test the common method biases of the data, and descriptive statistical analysis was carried out to investigate the average score of each variable. Next, correlation and regression analyses were conducted to investigate the relationship between neuroticism, ER (including cognitive reappraisal and expressive suppression), depression, and NSSI behavior. Then, the significance of mediating effects was tested with the SPSS plug-in Process 3.0 compiled by Hays.

## 3. Results

### 3.1. Common Method Biases Test

Harman’s single-factor test was performed to test the common method biases. All items of each questionnaire were taken as the entries of exploratory factor analysis, and the results showed that the first factor only explained 14.63% of the bias, lower than the critical standard of 40%, indicating that there was no serious common method bias in this study [30].

### 3.2. Gender Differences in Neuroticism, Cognitive Reappraisal, Expressive Suppression, Depression, and NSSI Behavior

Results suggested that gender differences were significant (*p* < 05). Female students scored higher than male students for all variables, including neuroticism, cognitive reappraisal, expressive suppression, depression, and NSSI behavior (Table 1).

### 3.3. Correlation Analysis of Neuroticism, Emotion Regulation (Cognitive Reappraisal/Expressive Suppression), Depression, and NSSI Behavior

The correlation analysis revealed that neuroticism was negatively associated with cognitive reappraisal/expressive suppression (*r* = −0.19, *p* < 0.05; *r* = −0.23, *p* < 0.05), but positively correlated with depression and NSSI behavior (*r* = 0.44, *p* < 0.05; *r* = 0.19, *p* < 0.05). Moreover, there were positive correlations between cognitive reappraisal and expressive suppression (*r* = 0.82, *p* < 0.05), and between depression and NSSI behavior (*r* = 0.32, *p* < 0.05). Expressive suppression was found to be negatively associated with depression and with NSSI behavior (*r* = −0.14, *p* < 0.05; *r* = −0.11, *p* < 0.05). However, no significant correlation was found between cognitive reappraisal and depression, or between cognitive reappraisal and NSSI behavior (*r* = −0.04, *p* > 0.05; *r* = −0.07, *p* > 0.05) (Table 1). It suggests that cognitive reappraisal is not significantly linked to depression or NSSI behavior in the presence of other variables, which did not affect the further regression and mediation analysis.

Furthermore, the correlation between neuroticism and NSSI behavior is non-significant in male students and significant in female students. While the correlation between emotion regulation (cognitive reappraisal/expressive suppression) and NSSI behavior is significant in male students and non-significant in female students (Table 1). The results demonstrate that gender differences exist for these variables, suggesting that they may also exist in the following analysis of regression effects and mediating effects between variables.

### 3.4. Effects of Neuroticism on NSSI Behavior: Mediating Effect of ER and Depression

Firstly, through collinearity diagnostic analysis, we found that the tolerance of each variable was 0–1 and *VIF* < 10, indicating that there was no multiple collinearity problem between variables. Then, regression analysis was performed to test the predictive effect of independent variables on dependent variables. After that, the significance of mediating effect was tested by SPSS plug-in Process 3.0 according to the bootstrap method (5000 samples were extracted). It is generally believed that gender, age, and grade have significant effects on several variables. Therefore, we made sure that gender, age, and grade were under control. Then, Process Model 6 was adopted for mediating effect analysis using neuroticism as the predictive variable, NSSI behavior as the outcome variable, and ER and depression as the mediating variables.

Table 2 displays the OLS regression models to test our research hypotheses. Regression analysis showed that neuroticism significantly negatively predicted cognitive reappraisal (*β* = −0.19, *p* < 0.01), however expressive suppression (*β* = −0.23, *p* < 0.01) significantly positively predicted depression (*β* = 0.44, *p* < 0.01). In addition, cognitive reappraisal was found to significantly positively predict depression (*β* = 0.26, *p* < 0.01), while expressive suppression significantly negatively predicted depression (*β* = −0.26, *p* < 0.01). Furthermore, depression significantly positively predicted NSSI behavior (*β* = 0.30, *p* < 0.01).

### 3.5. Mediating Effect

Mediation analysis showed that cognitive reappraisal, expressive suppression, and depression had significant mediating effects, with a mediating effect value of 0.041, accounting for 81.07% of the total effect. The mediating effect consists of the following four pathways of indirect effects: (1) pathway 1 neuroticism→depression→NSSI behavior; (2) pathway 2 neuroticism→cognitive reappraisal→depression→NSSI behavior; (3) pathway 3 neuroticism→expressive suppression→depression→NSSI; (4) and pathway 4 neuroticism→cognitive reappraisal→expressive suppression→depression→NSSI behavior. Among these four pathways, none of their confidence intervals (CIs) contained zero value, indicating that their indirect effect was significant (Table 3 and Figure 1).

### 3.6. Gender Differences Reflected in the Multivariate Regression Analysis of Neuroticism, Emotion Regulation (ER), Depression, and NSSI

Regression analysis showed that male college students’ neuroticism significantly negatively predicted cognitive reappraisal (*β* = −0.19, *p* < 0.01) and expressive suppression (*β* = −0.25, *p* < 0.001); however, it significantly positively predicted depression (*β* = 0.45, *p* < 0.001). In addition, expressive suppression significantly negatively predicted depression (*β* = −0.25, *p* < 0.05) (Table 4).

In comparison, the analysis showed that for female students, neuroticism significantly negatively predicted cognitive reappraisal (*β* = −0.26, *p* < 0.001) and expressive suppression (*β* = −0.25, *p* < 0.001); however, it significantly positively predicted depression (*β* = 0.39, *p* < 0.001). Cognitive reappraisal significantly positively predicted depression (*β* = 0.24, *p* < 0.05), while expressive suppression significantly negatively predicted depression (*β* = −0.27, *p* < 0.01). Moreover, depression significantly positively predicted NSSI behavior (*β* = 0.36, *p* < 0.001) (Table 5).

### 3.7. Gender Differences Reflected in the Mediation Effect Analysis of Neuroticism, Emotion Regulation, Depression, and NSSI

Mediation effect analysis demonstrated significant gender differences in the mediation pathways between neuroticism, emotion regulation, depression, and NSSI behavior. For male students, neuroticism mediates NSSI behavior primarily through expressive suppression→depression or depression. While for female students, neuroticism mediates NSSI behavior primarily through cognitive reappraisal→depression or depression (Table 6 and Table 7, Figure 2 and Figure 3).

## 4. Discussion

We found that neuroticism could significantly negatively predict ER strategies and positively predict depression and NSSI behavior, which is consistent with the findings of Nudelman et al. [31] and Shukla et al. [32]. High neuroticism and self-awareness means emotional instability and poor emotional regulation, which is related to a high level of depression and anxiety. According to the novel integrative cognitive model of depression put forward by Villalobos et al. [33], when individuals pay attention to, remember, and explain external stimuli, they tend to process external information negatively due to negative processing bias. Highly neurotic individuals often have low ER ability and are not good at seeking help from the outside world, so they are prone to increased depression. The interplay between mood-congruent cognitive control difficulties, cognitive biases, and rumination may ultimately lead to ineffective emotion-regulation strategies to downregulate negative mood and upregulate positive mood. Therefore, if negative emotions cannot be alleviated for a long time, highly neurotic individuals may avoid or alleviate unpleasant and disgusting emotional experiences to a certain extent by resorting to NSSI behavior. This strengthens the correlation between negative emotions and NSSI behavior, resulting in individuals’ dependency on NSSI behavior when processing negative emotions [34]. In addition, we found that cognitive reappraisal was not significantly linked to depression or NSSI behavior in the presence of other variables, especially expressive suppression. Therefore, it can be inferred that cognitive reappraisal may be associated with depression and NSSI behavior through the indirect mediating effect of expressive suppression.

The mediating effect results show that neuroticism does not directly affect NSSI behavior among college students, but through the mediating effect of depression. This confirms that negative emotional experience caused by external traumatic events is the proximal and key factor of NSSI behavior. Depression, anxiety, and stress each exert a direct effect on NSSI, which is mediated by cognitive reappraisal and expressive suppression [35]. After experiencing childhood maltreatment—such as sexual abuse, physical abuse and neglect; and emotional abuse and neglect [36]—highly neurotic individuals are more likely to have negative emotions such as depression, anxiety, guilt, and fear. If they fail to mobilize more cognitive resources and physiological energy to regulate negative emotions, rumination moderates the relationships between reactivity, intensity, and perseveration of emotion and NSSI, leading to a higher tendency towards NSSI behavior [37,38]. We, therefore, suggest that reasonably addressing long-term negative emotions, which can alleviate depression, may be an effective intervention to reduce NSSI behavior of college students.

Results of the mediating effect analysis also demonstrated that ER (cognitive reappraisal and expressive suppression) and depression play multiple mediating roles in the influence of neuroticism on NSSI behavior. The results showed that cognitive reappraisal and expressive suppression can effectively weaken the influence of neuroticism on depression, so as to reduce NSSI behavior. Some commonality was found in ER deficits across NSSI and suicidal ideation. That is to say, effective ER strategies (i.e., cognitive reappraisal and expressive suppression) can reduce depressive emotion, and thus diminish NSSI behavior. Such a finding provides an intervention pathway to decrease individuals’ NSSI behavior from the perspective of emotional regulation. Therefore, prevention and intervention efforts should be focused on teaching ER strategies to lower self-injury risk. The Internet may be used as an effective pathway to prevent NSSI, as there are plenty of specific emotion regulation strategies available on the Internet (such as relaxation, MBCT eight-week mindfulness training, acceptance, and expressive suppression) for people to learn, so as to reduce depression and NSSI behavior. A previous study found that through ER training, adolescents’ emotional instability decreased; depression was relieved; ER ability improved; and NSSI behavior, suicidal ideation, and suicidal behavior decreased significantly [39]. The reason may be that cognitive reappraisal emphasizes an objective perspective towards the situation inducing emotion, which helps to infiltrate more social cognition and rational thinking into the process of ER, and to enhance the rational regulation mechanism of the interaction between individual and social environment, thus alleviating depression and reducing the occurrence of NSSI behavior. Additionally, expressive suppression can reduce individuals’ direct perception of negative emotions, inhibit the endless interference of negative automatic thinking, avoid absolutization and terrible mood, and restore emotional equilibrium quickly. Therefore, both cognitive reappraisal and expression inhibition may be efficient strategies for individual ER, which can effectively alleviate depression and reduce NSSI behavior.

However, this is inconsistent with the findings of Gross et al. that expressive suppression is not an effective ER strategy, because it causes stronger physiological response instead of mitigating the internal negative emotional experience [20]. The reason for this discrepancy may be due to the dual-process model of ER in which expressive suppression can be overt and introvert [40]. Expressive suppression proposed by Gross was overt, which requires the initiation of conscious efforts. During this process, although behavioral expression of upcoming or ongoing emotions are suppressed, the psychological experience of and physiological response to emotions are enhanced. However, in the present study, we used a questionnaire survey to measure the ER strategies habitually adopted by college students over a long time. It is an introvert and automatic expressive suppression without conscious participation, and the psychological experience of and physiological response to emotion will not be enhanced. These findings demonstrate that expressive suppression is also an effective ER strategy.

We also found significant gender differences in the mediation effect of neuroticism, emotion regulation, depression, and NSSI: For male students, neuroticism mediates NSSI behavior primarily through expressive suppression→depression, or depression. While for female students, neuroticism mediates NSSI behavior primarily through cognitive reappraisal→depression, or depression. That is to say, when encountering negative life events, neurotic male students are not good at expressing their inner feelings and tend to suppress their emotions (expressive suppression), which may cause long-term depression, anxiety, or other negative emotions and eventually lead to NSSI behavior; while neurotic female students are prone to negative automatic thinking and are incapable of cognitive adjustment or restructuring (cognitive reappraisal), which may also generate negative emotions like depression and eventually lead to NSSI behavior.

Therefore, this study provides important evidence of the mediating effect of ER and depression on neuroticism and NSSI behavior and gender differences in this respect, which offers a psychological intervention pathway of NSSI behavior in clinics for men and women.

## 5. Limitations and Future Studies

First of all, this study only enrolled college students, without including younger teenagers and elder young adults who are also high-risk populations, which limited the interpretation of the results. Therefore, in further studies, age factors can be included in the analysis to investigate the relationship between personality traits, ER, depression, and NSSI behavior at different age levels [5] (junior middle school group, senior high school group, and university group). Secondly, studies on the influence factors of NSSI behavior are mostly cross-sectional, ignoring the role of time, which fails to thoroughly reveal the dynamic relationship of mediating pathways over time. A longitudinal follow-up study can be conducted to comprehensively investigate the changes in the development of the relationship between variables by incorporating a time factor [41]. Recent developments in statistical methods for analyzing longitudinal data provide efficient estimates of change and predictors of change over time [42]. Therefore, future research can adopt a continuous multiple tracking design to investigate the dynamic relationship between personality traits, ER, depression, and NSSI behavior over time. Furthermore, considering the correlation between NSSI behavior and suicidal ideation, we will explore the incidence of suicidal ideation among self-injuring individuals and its association with NSSI behavior in our future studies.

## 6. Conclusions

In conclusion, based on the general strain theory and ER Theory, this study measured variables of neuroticism, ER, depression, and NSSI behavior, and explored the mediating effect of ER and depression on neuroticism and NSSI behavior. We found that neuroticism among college students can significantly negatively predict ER strategies and positively predict depression and NSSI behavior; at the same time, ER and depression play a multiple chain mediating role in the influence of neuroticism on NSSI behavior.

## Figures and Tables

**Figure 1 ijerph-19-02885-f001:**
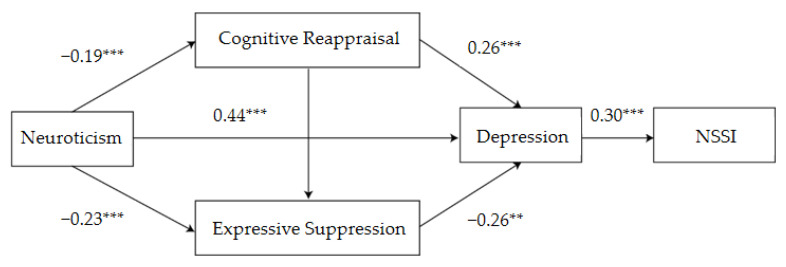
Influence of neuroticism on NSSI behavior: schematic diagram of the mediating role of emotion regulation and depression. ** *p* < 0.01, *** *p* < 0.001.

**Figure 2 ijerph-19-02885-f002:**
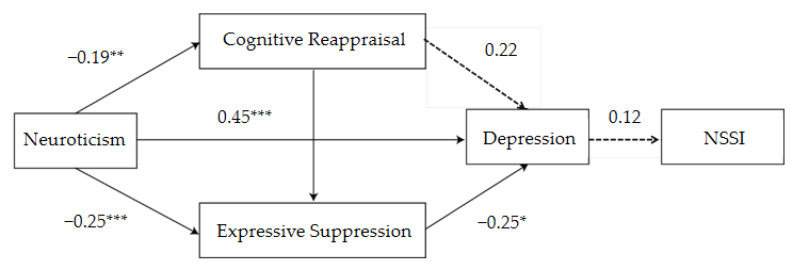
Influence of neuroticism on NSSI behavior: schematic diagram of the mediating role of emotion regulation and depression among male college students. * *p* < 0.05, ** *p* < 0.01, *** *p* < 0.001.

**Figure 3 ijerph-19-02885-f003:**
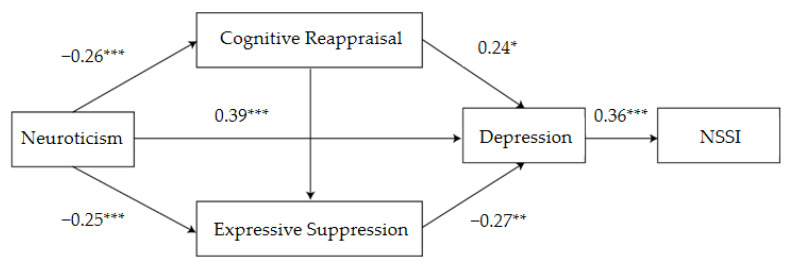
Influence of neuroticism on NSSI behavior: schematic diagram of the mediating role of emotion regulation and depression among female college students. * *p* < 0.05, ** *p* < 0.01, *** *p* < 0.001.

**Table 1 ijerph-19-02885-t001:** Gender differences in variables and the correlation between them.

Variables	Gender	M ± SD	1	2	3	4	5
Neuroticism	male	16.33 (± 3.64)	1				
	female	17.78 (± 3.40)	1				
	total	17.13 (± 3.58)	1				
2. Cognitive reappraisal	male	26.98 (± 6.20)	−0.19 **	1			
	female	28.36 (± 5.02)	−0.26 **	1			
	total	27.74 (± 5.61)	−0.19 **	1			
3. Expressive suppression	male	19.26 (± 4.17)	−0.25 **	0.85 **	1		
	female	19.85 (± 3.25)	−0.25 **	0.80 **	1		
	total	19.58 (± 3.70)	−0.23 **	0.82 **	1		
4. Depression	male	49.09 (± 9.10)	0.45 **	−0.07	−0.16 *	1	
	female	53.75 (± 9.00)	0.39 **	−0.07	−0.17 **	1	
	total	51.66 (± 9.33)	0.44 **	−0.04	−0.14 **	1	
5. NSSI behavior	male	0.79 (± 3.60)	0.09	−0.14 *	−0.18 **	0.32 **	1
	female	2.47 (± 5.61)	0.20 **	−0.08	−0.10	0.38 **	1
	total	1.71 (± 4.88)	0.19 **	−0.07	−0.11 *	0.32 **	1

Note: * *p* < 0.05, ** *p* < 0.01.

**Table 2 ijerph-19-02885-t002:** Multivariate regression analysis of neuroticism, emotion regulation (ER), depression, and NSSI.

Independent Variables	Dependent Variables	*R*	*R* ^2^	*F*	*β*	*t*
Neuroticism	Cognitive Reappraisal	0.19	0.04	17.36	−0.19	−4.17 ***
	Expressive Suppression	0.23	0.05	24.03	−0.23	−4.90 ***
	Depression	0.44	0.20	42.11	0.44	10.50 ***
Neuroticism	Depression	0.47	0.42	42.07	0.44	10.18 ***
Cognitive Reappraisal					0.26	3.51 ***
Expressive Suppression					−0.26	−3.43 **
Neuroticism	NSSI	0.33	0.10	13.58	0.04	0.8
Cognitive Reappraisal					−0.33	−0.03
Expressive Suppression					−0.04	−0.43
Depression					0.3	5.86 ***

Note: ** *p* < 0.01, *** *p* < 0.001.

**Table 3 ijerph-19-02885-t003:** Mediation effect analysis of neuroticism, emotion regulation, depression, and NSSI.

Pathways	ME	ER	95% CI	
			LL	UL
Neuroticism→Depression→NSSI	0.037	72.78%	0.021	0.060
Neuroticism→Cognitive Reappraisal→Depression→NSSI	−0.004	8.28%	−0.010	−0.001
Neuroticism→Expressive Suppression→Depression→NSSI	0.001	2.76%	0.000	0.004
Neuroticism→Cognitive Reappraisal→Expressive Suppression→Depression→NSSI	0.003	6.51%	0.001	0.008
Total indirect effect	0.041	81.07%	0.025	0.063

Note: ME: mediating effect; ER: effect ratio; CI: confidence interval; LL: lower level; UL: upper level.

**Table 4 ijerph-19-02885-t004:** Multivariate regression analysis of neuroticism, emotion regulation (ER), depression, and NSSI among male college students.

Independent Variables	Dependent Variables	*R*	*R* ^2^	*F*	*β*	*t*
Neuroticism	Cognitive Reappraisal	0.19	0.03	7.49	−0.19	−2.74 **
	Expressive Suppression	0.25	0.06	13.07	−0.25	−3.62 ***
	Depression	0.45	0.20	5.51	0.45	7.11 ***
Neuroticism	Depression	0.47	0.21	18.47	0.43	6.63 ***
Cognitive Reappraisal					0.22	1.88
Expressive Suppression					−0.25	−2.06 *
Neuroticism	NSSI	0.22	0.03	2.44	−0.01	−0.13
Cognitive Reappraisal					0.01	0.09
Expressive Suppression					−0.18	−1.32
Depression					0.12	1.54

Note: * *p* < 0.05, ** *p* < 0.01, *** *p* < 0.001.

**Table 5 ijerph-19-02885-t005:** Multivariate regression analysis of neuroticism, emotion regulation (ER), depression, and NSSI among female college students.

Independent Variables	Dependent Variables	*R*	*R* ^2^	*F*	*β*	*t*
Neuroticism	Cognitive Reappraisal	0.26	0.07	18.19	−0.26	−4.26 ***
	Expressive Suppression	0.25	0.06	16.39	−0.25	−4.05 ***
	Depression	0.39	0.15	43.07	0.39	6.56 ***
Neuroticism	Depression	0.42	0.17	17.34	0.38	6.34 ***
Cognitive Reappraisal					0.24	2.49 *
Expressive Suppression					−0.27	−2.76 **
Neuroticism	NSSI	0.39	0.14	1.70	0.06	0.86
Cognitive Reappraisal					−0.05	−0.52
Expressive Suppression					0.02	0.20
Depression					0.36	5.50 ***

Note: * *p* < 0.05, ** *p* < 0.01, *** *p* < 0.001.

**Table 6 ijerph-19-02885-t006:** Mediation effect analysis of neuroticism, emotion regulation, depression, and NSSI among male college students.

Pathways	ME	ER	95% CI	
			LL	UL
Neuroticism→Depression→NSSI	0.010	100%	0.001	0.026
Neuroticism→Expressive Suppression→Depression→NSSI	0.0004	100%	0	0.003
Neuroticism→Cognitive Reappraisal→Expressive Suppression→Depression→NSSI	0.001	0.04%	0	0.004
Total indirect effect	0.018	100%	0.005	0.041

Note: ME: mediating effect; ER: effect ratio; CI: confidence interval; LL: lower level; UL: upper level.

**Table 7 ijerph-19-02885-t007:** Mediation effect analysis of neuroticism, emotion regulation, depression, and NSSI among female college students.

Pathways	ME	ER	95% CI	
			LL	UL
Neuroticism→Depression→NSSI	0.047	730.66%	0.021	0.087
Neuroticism→Cognitive Reappraisal→Depression→NSSI	−0.007	770.66%	−0.021	−0.001
Neuroticism→Cognitive Reappraisal→Expressive Suppression→Depression→NSSI	0.007	290.24%	0.002	0.018
Total indirect effect	0.051	750.15%	0.024	0.089

Note: ME: mediating effect; ER: effect ratio; CI: confidence interval; LL: lower level; UL: upper level.

## Data Availability

The datasets acquired and/or analyzed during the current study are available from the corresponding author on reasonable request.

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
