# Peer review of "The Relation between Neuroticism and Non-Suicidal Self-Injury Behavior among College Students: Multiple Mediating Effects of Emotion Regulation and Depression"

_ijerph, 2022, doi:10.3390/ijerph19052885_

Round 1
Reviewer 1 Report
The title is clear and describes the content of the manuscript.
The summary also complies with the corresponding sections.
In the introduction and justification, the background of the problem is clearly stated, as well as those issues that are not clear or require resolution.
It is recommended to include the objectives of the work before the hypotheses.
As for the results, they are well structured, synthesizing extensive information that is well organized. However, it would be advisable not to include 0 before the decimal (p<0.05).
Better organize the discussion and conclusions. It is recommended to modify the statement (line 220-221): “Results of regression analysis showed that neuroticism could significantly negatively predict ER strategies and positively predict depression” (this information is already included in the results section). The information would be better organized if the study findings were presented in connection with the objectives and hypotheses.
On the other hand, include a section with the contributions of the work and its applicability so that the information found on lines 259-260 can be included: “Therefore, prevention and intervention efforts should be focused on teaching ER strategies to lower self- injury risk”.
In the Limitations section, these have not been included.
The information included in the section “Limitations and future lines”, which ranges from lines 287 to 292 (“Based on the General Strain Theory and ER Theory, this study measured variables of neuroticism, ER, depression and NSSI behavior, and explored the mediating effect of ER and depression on neuroticism and NSSI behavior.We found that neuroticism among college students can significantly negatively predict ER strategies and positively predict depression and NSSI behavior;at the same time, ER and depression play a multiple chain mediating role in the influence of neuroticism on NSSI behavior.”), seem like conclusions.
The information that goes from Line 293 to 297 (“Although personality traits are stable in a sense, they also have certain development such plasticity, and individuals' ER ability will become more mature with age and de velopment of social cognition. The predictive relationship between personality traits and NSSI behavior and the mediating effect of ER and depression may also have a certain stage of development.”), is not very clear in that section.
Reviewer 2 Report
This is well-written but somewhat frustrating paper.
First, I am puzzled as to why the researchers have not analysed data separately by gender (male and female). If there are differences in the explanatory models, this would be interesting, and perhaps of clinical significance. I am not convinced, at this stage, that the NSSI scale measures an important dimension of behaviour (I tried to obtain the Wan et al., 2018 study on the scale, but this journal was not available to me). Why should the tendency to print/tattoo letters or symbols on skin, to pinch or lightly cut oneself be of clinical significance, without any signs of suicidal ideation. As it stands, NSSI appears to be a sub-facet of neurotic depression, and is not an important or the main focus for clinical intervention.
This study should have included a measure of suicidal ideation - questions of thoughts about self-destruction are included, for example, in the Beck depression scale, and there are special scales for this purpose, which could have been used in the present study. The present research could be offered as a 'pilot study', given the relatively small N, and a fuller study would explore whether NSSI and Suicidal Ideation are in fact correlated, or predictive of one another. Since body alteration may have phenomenological differences between genders, data for males and females should be analysed separately e.g. in a replication study, is anorexia/bulemia a form of NSSI?
My judgment is that the incidence figures for NSSI given in the introduction are too high - see Matera (2021) in the MDPI journal Brain Sciences, who give figures of between 1 and 7%.
Matera, E., Margari, M., Serra, M., Petruzzelli, M. G., Gabellone, A., Piarulli, F. M., ... & Margari, A. (2021). Non-suicidal self-injury: an observational study in a sample of adolescents and young adults. Brain sciences, 11(8), 974.
Round 2
Reviewer 2 Report
This revision has carefully addressed the points made in my earlier review. I can now recommend publication without the need for any further changes.